# Fruit scent and observer colour vision shape food-selection strategies in wild capuchin monkeys

Amanda D. Melin [1,2], Omer Nevo [3], Mika Shirasu [4,5], Rachel E. Williamson [1], Eva C. Garrett [6], Mizuki Endo[7], Kodama Sakurai[7], Yuka Matsushita[7], Kazushige Touhara [4,5] & Shoji Kawamura [7]

The senses play critical roles in helping animals evaluate foods, including fruits that can change both in colour and scent during ripening to attract frugivores. Although numerous studies have assessed the impact of colour on fruit selection, comparatively little is known about fruit scent and how olfactory and visual data are integrated during foraging. We combine 25 months of behavioural data on 75 wild, white-faced capuchins (*Cebus imitator*) with measurements of fruit colours and scents from 18 dietary plant species. We show that frequency of fruit-directed olfactory behaviour is positively correlated with increases in the volume of fruit odours produced during ripening. Monkeys with red–green colour blindness sniffed fruits more often, indicating that increased reliance on olfaction is a behavioural strategy that mitigates decreased capacity to detect red–green colour contrast. These results demonstrate a complex interaction among fruit traits, sensory capacities and foraging strategies, which help explain variation in primate behaviour.

[1] Department of Anthropology and Archaeology, University of Calgary, Calgary AB T2N 1N4, Canada. [2] Department of Medical Genetics and Alberta Children's Hospital Research Institute, Cumming School of Medicine, University of Calgary, Calgary AB T2N 4N1, Canada. [3] Institute of Evolutionary Ecology and Conservation Genomics, University of Ulm, Albert-Einstein-Allee 11, 89081 Ulm, Germany. [4] Department of Applied Biological Chemistry, Graduate School of Agricultural and Life Sciences, The University of Tokyo, Bunkyo-ku, Tokyo 113-8657, Japan. [5] ERATO Touhara Chemosensory Signal Project, JST, The University of Tokyo, Tokyo 113-8657, Japan. [6] Department of Anthropology, Boston University, Boston MA 02215, USA. [7] Department of Integrated Biosciences, Graduate School of Frontier Sciences, The University of Tokyo, Kashiwa Chiba 277-8562, Japan. Correspondence and requests for materials should be addressed to A.D.M. (email: amanda.melin@ucalgary.ca) or to S.K. (email: kawamura@edu.k.u-tokyo.ac.jp)

Animal senses play a crucial role in facilitating both detection and evaluation of vegetative foods, and their evolution may be shaped by plant properties[1–3]. In turn, the properties of fruits and flowers, including their colours and scents, have in many cases evolved to be more discriminable to pollinators and seed dispersers[4,5]. For example, visual conspicuity of fruits against a leaf background increases with heightened chromatic and luminance contrast and may communicate information regarding nutritional content[6–9]. Similarly, fruit scent—the bouquet of emitted volatile organic compounds (VOCs)—varies with ripeness and is used by frugivores to select ripe fruit[10–12].

Inputs from different senses may influence the overall perception of stimuli by foragers[13,14]; e.g., visual signals may reinforce or suppress inputs from olfactory stimuli in passerine birds and in humans[15,16]. The reliance of frugivores on different characteristics of fruits and, in turn, the selection pressures they exert on fruit traits as seed dispersers strongly depend on the ability to perceive and integrate information from different sensory systems. When multiple signals are present for frugivores, signals may be redundant, reinforcing, differently informative, or informative at different distances or scales. Multimodal signals and their perception have been well studied in insect–flower interactions (e.g., see refs. [17–19]). However, with few exceptions[20–23], studies of sensory investigation by vertebrate frugivores have tended to focus on single sensory modalities[5,11,24,25]. As a result, even as it becomes clear that visual, olfactory and other senses play a role in mediating frugivore–plant interactions, the relative importance of each, and thus the potential for selection pressures on fruit traits, remain largely unknown.

Primates are important seed dispersers in many tropical ecosystems[26] and consume fruits that range considerably in size, colour, scent, mechanical protection and nutritive content[27–30]. Unlike other mammals, but akin to birds, many primates possess red–green colour vision[31–34] and rely on their visual sense to find and assess fruits[35,36]. Several studies have assessed the importance of colour vision for frugivory[37,38] and monkeys possessing red–green colour vision have higher feeding efficiencies for reddish foods than do dichromatic (colourblind) conspecifics[24,39]. Primates also show high olfactory sensitivity to and discrimination capacity for VOCs common in ripe wild fruits[5,11,12], yet data on fruit olfactory signalling and use of olfactory stimuli by wild primates are limited[3,40,41]. Moreover, as most recent research has examined olfaction in isolation, it remains unclear whether changes in fruit scent are also associated with visual cues and signals (fruit colour), and reward (nutrient content), as it may be costly for plants to invest in numerous signals[42].

Animals with intraspecific variation in sensory ability are particularly useful for teasing apart the role of different sensory stimuli in foraging decisions[43,44]. Due to naturally occurring colour vision variation in monkeys, the role of red–green colour contrast in fruit selection can be assessed by comparing the foraging strategies of dichromatic (red–green colourblind) and trichromatic (colour 'normal' relative to most humans) individuals foraging for ripe fruits[24]. Here we examine how variation in fruit colour and scent, as well as in the colour vision capacity of foragers, drive food-selection behaviour by wild capuchin monkeys (Cebus imitator). We focus on variation in sniffing behaviour as a proxy for the reliance on olfactory cues to ask the following questions: (1) How much variation in sniffing behaviour is present across fruit species? We predict that if fruit scent is informative, capuchins will use sniffing behaviours more often when selecting fruits that: (a) shift the composition of VOCs with ripening, (b) shift the overall amount of VOCs emitted with ripening. (2) Does fruit colour and monkey colour vision type impact the frequency of sniffing behaviour? We predict that fruits that do not shift in colour with ripeness will be investigated with sniffing, whereas fruits with a perceptible change in colour with ripeness will be sniffed less often. We additionally predict that dichromatic individuals will sniff fruits more often than trichromatic monkeys, especially when foraging on fruit with a perceptible red–green contrast from unripe fruit. (3) Do plants invest in visual signalling at the expense of olfactory signalling? If signal substitution minimizes cost expenditure of plants, we predict that fruits with a large change in colour contrast will have a small change in odour with ripeness and vice versa. If the two modalities reinforce signal efficacy or provide redundant information, we predict they will be positively correlated, i.e., we predict fruits with a larger colour change will also have a larger change in scent between ripe and unripe fruits[45]. We find extensive variation in use of olfaction by capuchin monkeys when assessing the fruits of different plant species; frequency of sniffing behaviours is positively correlated with increases in the volume of fruit odorants during ripening. Changes in fruit colour do not correlate with changes in fruit odour during ripening, nor with frequency of olfactory investigation. However, red–green colourblind monkeys sniff fruits more often than colour 'normal' conspecifics, indicating that animals with different colour vision types pursue different behavioural strategies to assess fruits. Taken together, our results demonstrate a complex interaction among fruit traits, sensory capacities and foraging strategies, and contribute to a growing understanding of the factors shaping plant signals and the sensory ecology of wild frugivorous primates.

## Results

**Importance of olfaction during fruit investigation.** We observed capuchins foraging on the fruits of 83 species (46,709 fruit investigation events). The frequency of sniffing behaviour varied substantially among plant species (Supplementary Fig. 1). For approximately two-thirds of the fruit species, sniffing behaviour was very rare or absent. For the remaining plant species, sniffing behaviour was more common, occurring in ~10–50% of all fruit investigation records. The breadth of this variation was also represented in the subset of 18 plant species (15,160 fruit investigation events) for which we quantified the colour and scent (Fig. 1).

**Impact of fruit colour and scent on foraging behaviour.** The general linear mixed model (glmm) that included fixed effects of colour vision phenotype, olfactory and visual conspicuity of ripe fruits was significantly better at explaining variation in sniffing behaviour than the null model, which included only random effects ($\chi^2$ (6) = 17.67, $p < 0.01$). Together, the fixed effects explained 45% of the overall variance; the full model (fixed and random effects) explained 91% of the variance in sniffing behaviour (Table 1). We found no significant main effect of chemical dissimilarity between unripe and ripe fruits on sniffing behaviour (glmm: $p > 0.05$; Fig. 2a). However, the overall scent increase ratio between unripe and ripe fruits was significantly positively correlated with an elevated occurrence of sniffing behaviours ($p = 0.02$). Plant species with a larger increase in the volume of odorants produced were sniffed significantly more often (Fig. 2b). No measure of fruit visual conspicuity (red–green contrast, blue–yellow contrast or luminance contrast) was significantly correlated with sniffing behaviour (glmm; $p > 0.05$).

**Impact of colour vision on sniffing behaviour.** Colour vision phenotype had a significant impact on sniffing behaviour. Controlling for fruit species, individual ID and the effect of fruit scent, dichromatic monkeys sniffed fruits more often than trichromatic monkeys did (38.8% vs. 28.6% of the data points, respectively; glmm: $p < 0.01$; Table 1; Fig. 3). As sex and colour vision type are confounded in this system, we also ran models for females only to verify that these effects are not driven by sex rather than colour vision phenotype. We achieved comparable results, i.e., a

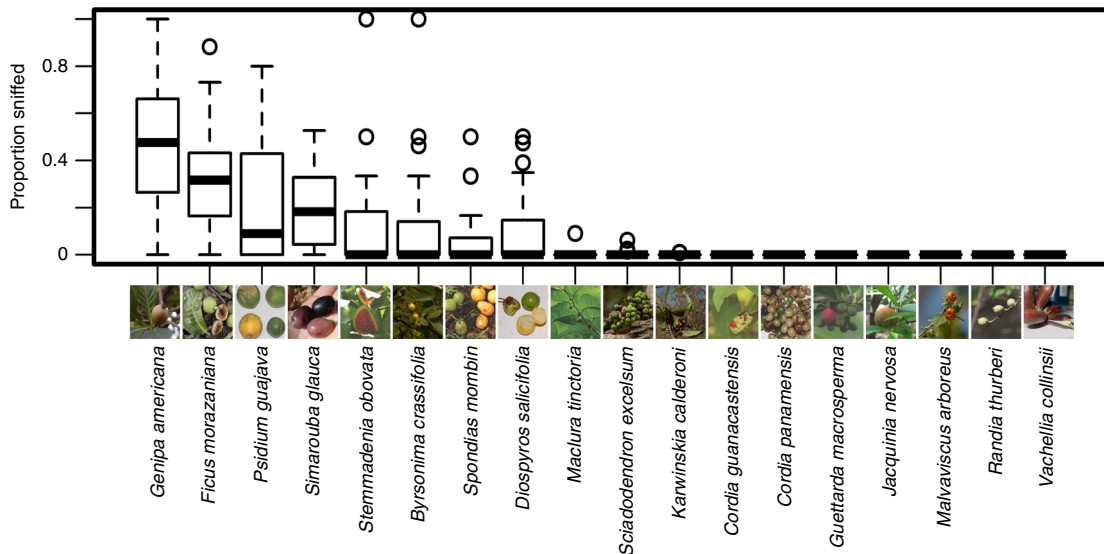

**Fig. 1** Variation in fruit sniffing behaviour by white-faced capuchins for 18 plant species. Box plots of fruit investigation sequences ($N = 15,160$) by individual white-faced capuchin monkeys that included at least 1 sniffing event. Medians (bar) are plotted along with upper and lower quartiles (box perimeters), and whiskers stretching to the first data point within 1.5 interquartile ranges of the box. Points beyond the whiskers are plotted as individual symbols. Source data are provided as a Source Data file. Photo credit for fruit images: AD Melin

| Table 1 Results of generalized linear mixed model | | | | |
|---|---|---|---|---|
| | **Estimate** | **SE** | **z-value** | **p-value** |
| Intercept | −1.02 | 4.51 | −0.23 | 0.82 |
| Colour vision type (trichromat vs. dichromat) | −1.07 | 0.37 | −2.9 | **0.004** |
| Scent increase ratio (log-transformed) | 1.07 | 0.46 | 2.31 | **0.02** |
| Chemical dissimilarity | −4.25 | 2.95 | −1.44 | 0.15 |
| Red–green distance (log-transformed) | −0.24 | 0.71 | −0.34 | 0.74 |
| Blue–yellow distance (log-transformed) | 0.7 | 1.03 | 0.68 | 0.49 |
| Luminance distance (log-transformed) | −0.76 | 0.94 | −0.81 | 0.42 |

Coefficients and *p*-values of all fixed effects. This model explained 91% of the variance in fruit sniffing behaviour by wild, white-faced capuchin monkeys; 45% of the variance was explained by the fixed factors shown in the table and 46% by the random factors (individual, plant species). *P*-values in bold are statistically significant at $p < 0.05$

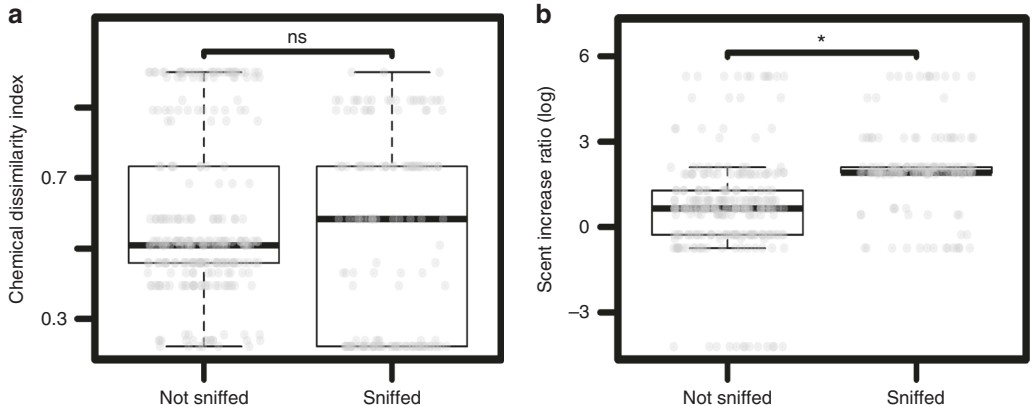

**Fig. 2** Ripe fruit olfactory conspicuity and monkey sniffing behaviour. **a** Chemical dissimilarity between conspecific ripe and unripe fruits calculated as Bray–Curtis dissimilarities based on the relative amounts of VOCs. **b** Scent increase ratio: total amount of scent emitted by an individual ripe fruit divided by the total amount of scent emitted by a conspecific unripe fruit. Each boxplot is overlaid on the raw data, jittered horizontally. Bars over boxes provide significance in a glmm (see Methods). ns, $p > 0.05$; *$p < 0.05$. Source data are provided as a Source Data file

significant effect of colour vision type (Supplementary Table 1). The impact of colour vision phenotype held irrespective of the red–green, blue–yellow or luminance contrast between ripe and unripe fruits; fruits ranging from high to low visual conspicuity along each of these dimensions were found in both the sniffed and non-sniffed categories (Fig. 3, Supplementary Figs. 2, 3). When we removed trichromatic females from the analysis to examine the impact of sex alone, we found that dichromatic

males sniffed fruits significantly less than dichromatic females did (Supplementary Table 2).

**Relationship between colour and scent among dietary fruits.** We found no or very weak correlation among any measure of visual contrast between ripe and unripe fruits with either the chemical

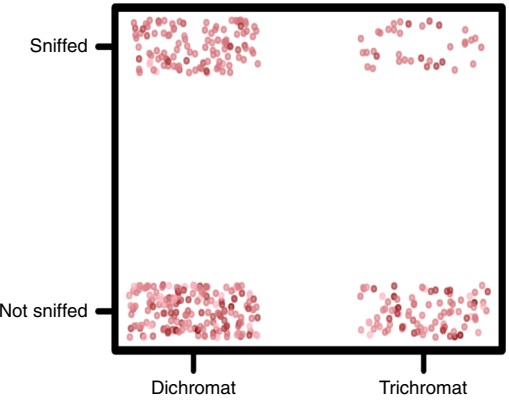

**Fig. 3** Sniffing behaviour and primate colour vision type plotted with red–green contrast between unripe and ripe fruits. Each dot represents an individual monkey–plant species combination. Data points were binomially classified as to whether that monkey was ever observed sniffing the fruits of that plant species and are vertically and horizontally jittered in each quadrant. Darker red dots represent a larger red–green chromatic contrast between ripe and unripe fruits in the visual space of a trichromatic capuchin monkey. Dichromatic monkeys sniffed fruits significantly more often than trichromatic monkeys did, regardless of red–green contrast, glmm: $p < 0.01$. Source data are provided as a Source Data file

dissimilarity or scent increase ratio (Fig. 4). In other words, difference in colour between ripe and unripe fruits of the same species was not related to the shift in scent volume or VOC composition with ripeness. This held for red–green chromatic contrast, blue–yellow chromatic contrast and luminance contrast. Colour and scent shifts with ripeness appeared to vary independently from each other.

## Discussion

Our prediction that capuchins will sniff fruits that produce greater volumes of VOCs when ripe was supported. We found the sniffing behaviour was positively correlated with an increase in emitted VOCs between ripe and unripe fruits. These results provide some of the first behavioural evidence of how wild frugivorous mammals use the chemical properties of food to inform their foraging decisions, and are consistent with recent research reporting that lemurs increase their reliance on olfaction when feeding on fruits whose scent changes with ripeness[5], and that primates are highly sensitive and attentive to some odorants[46].

Unlike the change in the amount of VOCs, the chemical dissimilarity between ripe and unripe fruits did not correlate with sniffing behaviour, in contrast to our prediction. The interpretation of this shift by capuchins may be poor[45], i.e., these monkeys may not be able to detect or are not attentive to the phenophase-specific bouquets of dietary fruits. However, evidence that chemical composition informs foraging decisions is present for spider monkeys, another neotropical primate[11], and for lemurs[5]. It is alternatively possible that capuchins do rely on differences in the chemical composition of fruit scent but that they particularly attend to subsets of VOCs. For example, primates may be particularly attracted to aliphatic esters[5,47] or ethanol[48,49], and bats tend to be attracted to monoterpenes[10]. Lacking a priori knowledge about which VOCs are used to identify ripeness, our measurement of ripe–unripe dissimilarity

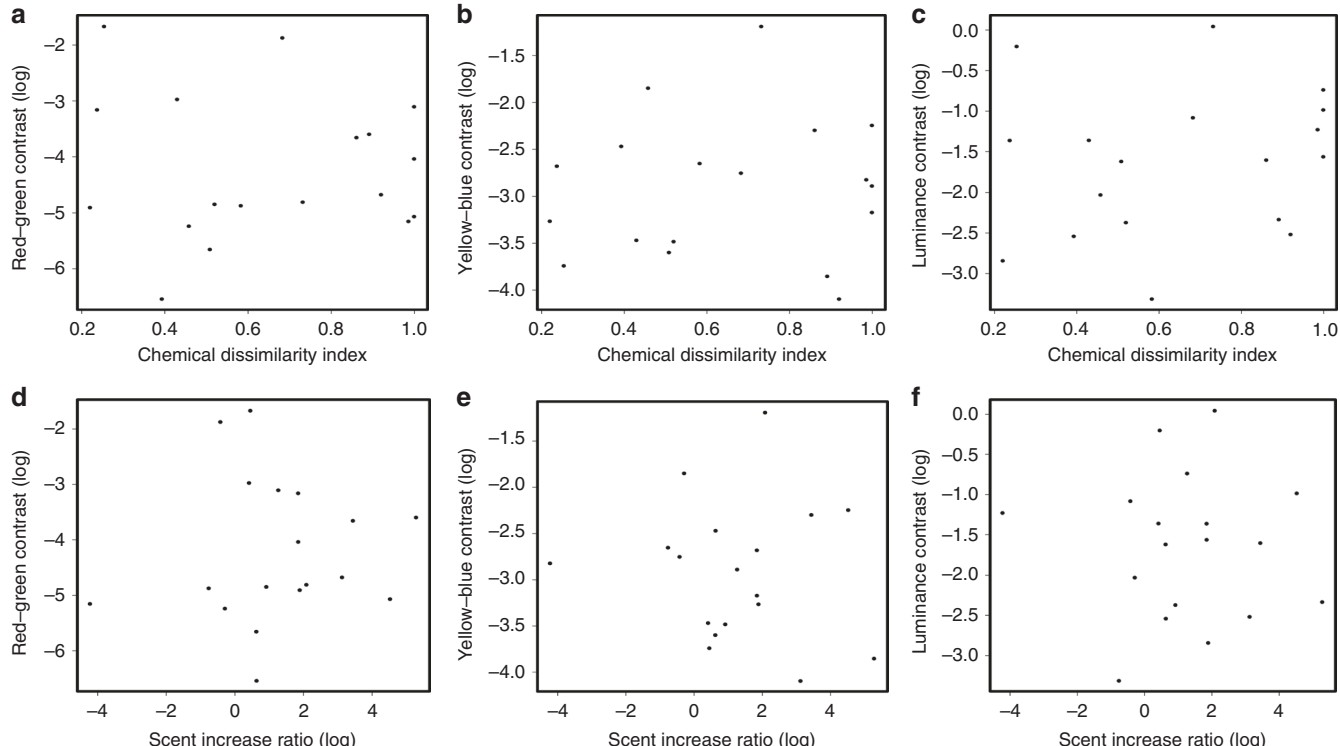

**Fig. 4** Correlation between fruit visual and olfactory changes with ripeness. Scatterplots showing three measures of colour distance between unripe and ripe fruits: **a** red–green contrast, **b** yellow–blue contrast, **c** luminance contrast against the Bray-Curtis chemical dissimilarity index, and against the log scent increase ratio (**d**–**f**). (Pearson's correlation, ripe–unripe chemical dissimilarity: −0.05, 0.07 and 0.17, respectively; scent increase ratio: 0.07, −0.1 and −0.04). Source data are provided as a Source Data file

(Bray–Curtis dissimilarity index) included all VOCs in the dataset. Given that some of those are likely to play other roles such as defence[50], this inclusive approach may have introduced noise (variation in compounds irrelevant to capuchins) that hindered us from detecting the effect of changes in chemical composition that are meaningful to capuchins. The olfactory receptor (OR) gene family is the largest and most diverse in the mammalian genome[51]. Capuchin monkeys, similar to other primates, have hundreds of functional receptors, yet little is known about which compounds they are sensitive to[36]. Future work that seeks to (1) de-orphan these ORs and (2) evaluate structural or reactive features of VOCs in association with sniffing and feeding behaviours may help us to link the chemical composition of VOCs to olfactory behaviours. This will permit a better understanding of how the content of fruit signals is acquired and interpreted by frugivores[52]. Further, it would allow future studies to focus on the VOCs that are more likely to be ecologically relevant and will thus be more likely to detect the effect of subtle chemical differences between ripe and unripe fruits on capuchin sensory behaviour.

Sniffing behaviour was not significantly affected by colour contrast in any dimension between the ripe and unripe fruits. Rather, capuchins sniffed some (but not all) fruits of species that underwent a conspicuous colour change as well as fruits that stayed green throughout ripening. This suggests that colour and scent signals are not redundant. Our conclusions differ somewhat from findings that olfactory investigation of fruits is negatively correlated with the luminance and blue–yellow contrasts between fruits and leaves in spider monkeys[44]. This may reflect differences in foraging strategies between the primate species or behaviours for the fruit species examined;[53] consistent with Hiramatsu et al.[44], we find that monkeys sniff visually cryptic, evergreen figs (*Ficus morazaniana*) relatively often (ca. 40% of the time; Supplementary Fig. 1). However, we also find many fruits that are sniffed that also have high colour contrast, suggesting correlation of visual and olfactory cues, and signals vary inconsistently among plants. This variation likely underlies complex selection pressures from seed dispersers. Furthermore, in fruits that undergo both a scent and a colour change, fruit scent may reinforce and potentially amplify the information contained in the other modality, as has been found in captive experiments[22]. It is alternatively possible that in these cases these modalities contain different information that serves different purposes. For example, colour changes may occur first and send a signal from a long distance to draw animals to the food patch[54,55], whereas fruit softness and scent may communicate whether an individual fruit is edible to promote optimal seed dispersal[5]. This may speak to the different challenges of (1) attracting frugivores and then (2) informing decisions regarding nutritional and mechanical suitability for ingestion. Likewise, animals may use different sensory systems to find food sources than those used to evaluate which particular foods to ingest, and this may vary by plant species. Future studies will benefit from investigating these and related ideas comprehensively to promote fundamental understanding of plant–frugivore interactions and natural features shaping animal sensory ecology.

Red–green colourblind (dichromatic) capuchins sniffed fruits more often than did trichromats, regardless of fruit colour, mirroring and extending previous results found for capuchins foraging on visually conspicuous and cryptic figs (*Ficus* sp.)[56]. Together, these lines of evidence point to an overarching behavioural strategy by dichromats that is more reliant on their sense of smell. As has been suggested to explain the enhanced tactile sensitivity of blind human Braille readers[57], it is possible that monkeys with a narrower chromatic visual experience (dichromats) have a more sensitive sense of smell through enhanced fidelity in the neural transmission of olfactory stimuli. The increased sniffing behaviour may reflect a

behavioural correlate of this. In addition, dichromatic monkeys may learn from a young age to rely on odour as their chromatic visual world is much narrower than trichromatic monkeys. Data on olfactory threshold sensitivity and cognitive correlates in visually impaired humans are mixed. Blind subjects have been found to have lower olfactory sensitivity thresholds, larger olfactory bulbs and to rely more on their sense of smell than sighted subjects[58]. For example, compared with matched-age sighted children, visually -impaired children used olfaction more often to glean social information and paid more attention to odours of unfamiliar foods[59]. However, a recent meta-analysis concluded that blindness seems not to affect odour identification, odour discrimination or odour thresholds systematically. Importantly, however, this review assessed olfactory performance and not relative use of the sense of smell during food selection or facets of daily life. Further work exploring the development of sensory biases and neurological correlates among primates with different colour vision types would be instructive[60].

In sum, our results begin to reveal how a complex web of variables—from fruit colour and odour to sensory capacities of foragers—influence food-selection strategies in a neotropical primate. We demonstrate the value of studying sensory ecology using a multimodal framework, in which the visual and olfactory signals and cues interact with sensory capacities of foragers. Study of other senses, including haptic perception and gustation, as well as other species that possess similar intraspecific variation in sensory capacities, will expand the scope and breadth of this fascinating area of enquiry. We anticipate that these studies will also contribute to debate on the evolution of fruits traits and interactions between plants and seed-dispersing animals.

## Methods

**Study site and species.** We studied four groups of white-faced capuchin monkeys (*C. imitator*) in Sector Santa Rosa (SSR), Área de Conservación, Guanacaste, in northwestern Costa Rica. SSR is situated in a tropical dry forest, with strong seasonal oscillations in temperature and rainfall[61,62]. We collected behavioural data over 25 months nearly equally split between the rainy (typically mid-May through mid-November annually) and dry seasons: May–July 2004, January–May 2005, January–May 2007, September 2007–January 2008, and May–September 2008, February 2009, June 2009 and August 2009. In SSR, capuchins consume the fruits of over 100 plant species and improve the germination of seeds for many of the dietary species[29,63,64]. Similar to most other platyrrhines, capuchins possess intraspecific colour vision variation due to X-linked variation of the *OPN1LW* opsin gene[31,34]. Females homozygous for the *OPN1LW* opsin allele and all males (hemizygous) are dichromatic and possess red–green colourblind vision. Heterozygous females have trichromatic vision, similar to colour-normal humans. The genotypes of all individuals were determined by sequencing the amino acids at opsin tuning sites from fecal DNA, following established methods and previously reported[65]. In total, we studied 75 individuals: 51 dichromats and 24 trichromats (Supplementary Data 1).

**Behavioural data collection.** We conducted focal animal sampling for up to 10 min duration[66]. We were flexible with the duration of the focal follow to accommodate challenges in maintaining clear views of face and hands at all times, and to sample as many monkeys as possible in the same tree to control for ecological conditions[67]. We recorded detailed food investigation sequences and noted whether the fruit was sniffed or not before being eaten or rejected. We discarded sequences for which we could not clearly see the handing of the fruit, including the final outcome. A two-observer system (one caller, one recorder) was used so that the observer did not look away from the focal monkey, even fleetingly, to record a behaviour. When the observer's view of the monkey was partially obstructed, 'out of sight' was coded until a clear view was again available.

**Fruit sampling.** Data on fruit colour and scent were collected for a subset, 18 species, of the dietary fruit in two field seasons as follows: (1) July–August 2012 and (2) December 2015–June 2016 (Supplementary Data 1). The choice to sample a subset was based on financial limitations and access to fruits across ripeness categories. We were strategic in our selection, choosing fruits spanning different colour categories and including fruit species for which monkey varied in their propensity for olfactory investigation, ranging from never sniffed to often sniffed. In field season 1, we measured fruits from the following species: *Byrsonima*

*crassifolia, Cordia guanacastensis, Cordia panamensis, Genipa americana, Guettarda macrosperma, Jacquinia nervosa, Maclura tinctoria, Malvaviscus arboreus, Psidium guajava, Randia thurberi, Spondias mombin, Stemmadenia obovata* and *Vachellia collinsii.* In field season 2, we measured fruits from the following species: *Diospyros salicifolia, Ficus morazaniana, Karwinskia calderoni, Sciadodendron excelsum* and *Simarouba glauca).* We measured several unripe and ripe fruits of each species in the same sampling period; numbers of fruits measured varied and depended on availability of fruits (Supplementary Table 3). We collected fruits directly from trees that monkeys fed in, selecting fruits that were free of obvious insect damage or rot, and measured them within 3 h of collection. We selected unripe fruits that were approximately the same size as ripe fruits, i.e., excluding the small, unripe specimens. For 14 of the 18 plant species, ripeness was judged by the palpable softening of fruits. We selected fruits from the ends of the ripeness distribution, i.e., clearly unripe and hard vs. clearly ripe and soft, excluding fruits that appeared transitional or in a 'mid-ripe' phenophase. For the remaining four species, two were covered with a hard husk (*P. guajava* and *J. nervosa*) and ripeness was judged by colour change. For two dehiscent species (*S. obovata* and *Vachellia collinsii*), fruits were considered ripe when the husk opened to expose the arillate seeds.

**Colour contrast models**. To quantify fruit colour, we measured the spectral reflectance of fruits on-site using a portable spectrophotometer (USB4000; Ocean Optics, Inc.) and LS-1 light source. The spectrometer was calibrated before each session of data collection, using a white reflectance standard (WS-1-SL). We took five measurements per item and calculated a mean reflectance value for ripe fruits and for unripe fruits for each species. We modelled three measures of colour contrast between ripe and unripe fruits based on the sensitivities of capuchin monkey cone photoreceptors: red–green contrast, blue–yellow contrast and luminance (brightness) contrast under a forest shade illuminant recorded in SSR. The red–green contrast would be visible only to trichromatic monkeys, whereas all colour vision types would have similar abilities to perceive blue–yellow and luminance contrast.

Illuminant data used in these models were absolute irradiance spectra recorded under forest shade in SSR, using a USB2000 spectrometer that was calibrated with a 3100 K tungsten halogen calibration light source (LS-1-CAL, Ocean Optics). The spectra were taken through a cosine corrector (CC-3, Ocean Optics) attached to an optical fibre (QP200-2-UV/VIS, Ocean Optics). We calculated the quantum catch of photons incident on each photoreceptor type in the retina according to Equation 1[68], where $Q_i$ represents the quantum catch of a photoreceptor $i$ over the range of the primate visual spectrum, $R(\lambda)$ represents the reflectance spectrum, $I(\lambda)$ represents the irradiance spectrum and $S_i(\lambda)$ is the spectral sensitivity function of the $i$-th photoreceptor. We modelled chromatic and achromatic (luminance) distances using the trichromatic phenotype with $\lambda_{\max}$ values of 426 (*S*), 532 (*M*) and 561 (*L*)[65]. The chromaticity of each item was modelled as a ratio of the quantum catches by different *L*, *M* and *S* cones: red–green chromaticity = $Q_L/(Q_L + Q_M)$; blue–yellow chromaticity = $Q_S/(Q_L + Q_M)$. The relative luminance value, $L + M$, of each object was estimated by dividing its $Q_L + Q_M$ value by that of a hypothetical white surface that reflects 100% of the illuminant

$$Qi = \int_{400}^{700} R(\lambda)I(\lambda)S_i(\lambda)d\lambda \qquad (1)$$

The visual contrast for each colour channel was defined as $\Delta f_i = |\ln(Q^r_i) - \ln(Q^u_i)|$, where $Q^r_i$ and $Q^u_i$ denote the quantum catches of the receptor$_i$ ($i$ = L, M, S) for ripe fruit and unripe fruit of the same species, respectively. The luminance contrast between ripe fruit and unripe fruit was give as $\Delta L = |\ln(Q^r_M + Q^r_L) - \ln(Q^u_M + Q^u_L)|$. The blue–yellow contrast was defined as BY = $|\Delta L - \Delta f_S|$. The red–green contrast was defined as RG = $|\Delta f_L - \Delta f_M|$[69,70]. All analyses were carried out in MATLAB R2017. The blue–yellow and luminance contrasts calculated with this trichromatic model are very similar to the corresponding values calculated with dichromatic models. Our overarching results remain unchanged whether we use the spectral sensitivities of opsins coded by other *L/M* allele combinations.

**Measuring fruit scent**. To quantify fruit scent, we sampled the VOCs passively (without a pump) from the fruits using MonoTrap RGC18 TD rods (GL Sciences, Inc.). These are portable scent traps designed for strong passive absorbance based on a graphite carbon and octa-decyl-silyl composition. We also included adsorbents incubated in empty bags for the purpose of identifying potential contaminants. Following VOC collection, each adsorbent was stored in 0.3 ml low absorption glass vials (SUPELCO, Bellefonte, USA) until they were shipped to the University of Tokyo. Our collection methods varied slightly between scent sampling periods, but unripe fruits and ripe fruits of the same species were always measured in the same sampling season.

In sampling period 1 (2012), when available, multiple fruits of the same ripeness category were put into one inert Tedlar bag (Tedlar Bag AA-1, GL Sciences, Inc., Tokyo, Japan), along with a MonoTrap RGC18 adsorbent for solvent extraction (GL Sciences, Inc., Tokyo, Japan). We made this choice, because we wanted to ensure sufficient fruit volume to detect the VOCs for the species/ripeness condition. Fruits were incubated with adsorbents at room temperature for 22 h. Following VOC collection, each adsorbent was stored in 0.3 ml low absorption glass vials (SUPELCO, Bellefonte, USA) in the dark at 4 °C, until they were shipped to the University of Tokyo. Subsequently, we confirmed that single-fruit sampling

was robust and representative through additional in-house experiments. In sampling period 2, each fruit was given its own inert bag (Toppits oven bag, Minden, Germany) and volatiles were adsorbed by MonoTrap RGC18 TD for thermal desorption (GL Sciences, Inc., Tokyo, Japan). When available, multiple samples were tested for each species. In field season 2, fruits were incubated with adsorbents at room temperature in a dark location for 2 h. Adsorbents were then in the dark at −20 °C, until they were shipped to the University of Tokyo. In both field seasons, we also included adsorbents incubated in empty bags for the purpose of identifying potential contaminants.

We standardized all measurements to either the average amount per one fruit in the analysis of overall amount, or set the overall scent to 1 when analyzing chemical dissimilarity between ripe and unripe fruits. We analysed all samples using benchtop gas chromatography-mass spectrometry (GC-MS). We changed our GC-MS machine between sampling seasons; however, because our analysis establishes ripe–unripe difference indices based only on the difference within species, and ripe fruit and unripe fruit of the same species are processed in the same field season using the same methods and materials, we do not expect this to bias our results.

VOC samples from field season 1 (MonoTrap RGC18) were measured with a benchtop GC-MS (Shimadzu GCMS-TQ8030; Shimadzu, Kyoto, Japan) with stabilwax column 60 m × 0.32 mm i.d. with a film thickness of 0.5 μm (Restek Corporation, PA, USA) combined with a sniffing port equipped with a Sniffer9000 system (Brechbuhler, Houston, TX) in splitless mode (MS and sniffing port at ratio of 1:3). We added 200 μL of dichloromethane and sonicated the samples for 5 min to extract volatile compounds from the adsorbents. We then injected 1 μl of the extracted dichloromethane solvent into the GC-MS injection port. The column temperature was programmed to rise at 15 °C/min from 50 °C (2.5 min hold) to 100 °C and then 12 °C/min from 100 °C to 230 °C (5.2 min hold). The interface temperature was maintained at 230 °C and the ion source temperature was maintained at 230 °C. We analysed all samples from sampling period 2 (MonoTrap RGC18 TD) using a benchtop GC-MS (Shimadzu GCMS-QP2010; Shimadzu, Kyoto, Japan) with stabilwax column 60 m × 0.32 mm i.d. with a film thickness of 0.5 μm (Restek Corporation, PA, USA) combined with a thermal desorption system OPTIC4 (GL Science, Tokyo, Japan). Samples were again run in splitless mode. The initial temperature of the cryo-trap module, regulated by OPTIC4, was set at −150 °C using liquid nitrogen. The cryo-trap module was then heated to 250 °C at a rate of 50 °C/s and held for 3 min to inject the trapped compound into the capillary columns of GC-MS. The temperature of the vaporization chamber of GC-MS, regulated by OPTIC4, was programmed to rise at 5 °C/s from 50 °C to 230 °C and the column temperature was programmed to rise at 5 °C/min from 50 °C (2.5 min hold) to 230 °C (5.5 min hold). The interface temperature was maintained at 230 °C and the ion source temperature was maintained at 230 °C.

For both sampling seasons, mass spectra were obtained in full scan mode (range: *m/z* 29–400) by electron ionization. We analysed the data using Analyzer Pro (SpectralWorks, Cheshire, UK); after we aligned the peaks, the area of each peak was calculated. Individual peaks were tentatively identified using the National Institute of Standards and Technology library database (NIST 14) using a >85% confidence criterion and given a unique ID. Although this method is not sufficient to accurately identify VOCs, it was sufficient to identify the compound class of most VOCs and to remove known contaminants such as phthalates and siloxanes. Importantly, the limitations in accurately identifying the compound names does not impact our results or conclusions. For each fruit species, we aligned the peaks so that a compound was always the same ID across samples. This means that the compound ID itself has no effect on the distance metric we calculated and hence does not influence the results. To reduce the amount of noise and focus on more relevant VOCs, we also excluded all very rare compounds, defined as those identified in <20% of the samples in each species. Our rationale is that if a compound is consistently present across samples, it is less likely to be a contaminant and more likely to be of biological relevance. The cutoff of 20%, while somewhat arbitrary, serves to strike a balance between erring on the side of being too conservative, i.e., excluding compounds that may be biologically relevant, or on the side of being too permissive, i.e., including compounds that are likely to be contaminants or present in biologically insignificant amounts. Importantly, the compounds that were excluded were present in very small amounts meaning that their effect, if included, on the indices we calculated and used for our analysis would have been extremely small.

**Olfactory conspicuity calculations**. We chose two indices that approximate the olfactory conspicuity of ripe fruits: (1) scent increase ratio, which measures the degree to which a species increases the amount of fruit scent upon ripening, and (2) chemical dissimilarity, which is a measure of how similar the chemical constituents are between unripe and ripe fruits. We chose these variables based on two criteria. First, both have been suggested to play a role in primate fruit selection in other systems[70]. Second, as both quantify the difference between ripe and unripe fruits within a single plant species (collected within a single field season), we expect they will be robust to the differences in the sample collection in the two datasets.

To calculate the scent increase ratio, we summed up the amount of all VOCs based on total ion chromatogram (TIC) area in each species and then divided the amount emitted from a single ripe fruit by the amount emitted from a single unripe fruit. As such, it is fully independent of the chemical composition of the scent bouquet. As large ripe fruits develop from large unripe fruits and vice versa, it is also independent of fruit size. To calculate ripe–unripe chemical dissimilarity, we converted the amount of VOCs to relative amounts by dividing

the volume of each VOC by the overall volume of VOCs in that species/ripeness level. This renders this index independent of fruit size and of the overall amount of VOCs. We then calculated the Bray–Curtis dissimilarity index between the ripe and unripe fruits of each species. This method quantifies the difference of frequency spectrum of chemical compounds between ripe and unripe fruits in a species. We chose to use Bray–Curtis dissimilarities, because, as opposed to other methods, they ignore compounds absent in both ripe and unripe fruits in a species, and thus avoid inflating the similarity between a pair of samples that both lack a certain VOC[71]. For species with more than one data point per condition (i.e., the samples from sampling period 2), we averaged the values to create an 'average unripe fruit' and an 'average ripe fruit' metric for each analysis.

A limitation of our study is that the accuracy of quantifying VOC TIC area would have been improved by using dose–response curves for the different compound classes. Unfortunately, this was not practical because of the very large and diverse dataset, which included hundreds of different VOCs across 18 fruit species, each of 2 ripeness states. However, we believe that the effect of this is minimal for our goals, because our index does not aim to assess absolute amounts but rather to calculate the ratio between the overall amount in ripe and unripe fruits within species. As the chemicals present in ripe and unripe fruits of a given species tend to come from the same VOC classes, any calibration factor would be negated to a large extent when calculating the ratio. Therefore, the index we use is a reasonable approximation for the difference in overall VOC emission between ripe and unripe fruits of a single species.

**Statistical analyses**. To assess whether plant and animal colour vision characteristics correlate with sniffing behaviour, we used a generalized linear mixed model. The response variable was a binomial variable noting whether a given monkey was ever observed sniffing fruits of a given plant species. We included the individual's colour vision phenotype (dichromatic or trichromatic) as a fixed factor, as well as the chemical dissimilarity index, scent increase ratio, red–green contrast, blue–yellow contrast and luminance contrast between ripe and unripe fruits of the same species. The latter four were log-transformed to meet the assumptions of the model. We also included plant species and monkey ID as random intercept factors. We conducted a logistic regression (logit link function) using lme4:glmer()[72]. To test that no assumptions were violated, we verified that random effects were normally distributed, calculated the dispersion parameter and verified that there are no collinearity issues using car:vif()[73]. We tested the significance of the full model by constructing a null model containing only the random factors and comparing them using a $\chi^2$-test. We ran two additional models to examine the potentially confounding effects of sex and colour vision type. (1) We first ran a model that included only females in the analyses to assess colour vision type in the absence of male data. (2) We then ran a model that included only dichromats (removing trichromatic females) to test for an effect of sex. The lower sample sizes of the latter two models caused convergence warnings. To remedy this, we simplified the model by removing the fruit colour categories, which successfully resolved the warning message. Importantly, the results were essentially the same whether or not fruit colour was included, i.e., no change in which variables had a significant effect. We report the results of the simpler models that converged well (Supplementary Table 1, 2). Results of the full and simplified models can be found in the source code. All analyses were conducted on R 3.4.3 (R Core Team 2014) and all statistical tests were two-sided.

**Ethics statement**. This research adhered to the laws of Costa Rica, the United States and Canada, and complied with protocols approved by the Área de Conservación Guanacaste and by the Canada Research Council for Animal Care through the University of Calgary's Life and Environmental Care Committee.

**Reporting summary**. Further information on research design is available in the Nature Research Reporting Summary linked to this article.

## Data availability
Primary foraging data, chromatic and luminance distance data between ripe and unripe fruits, and chemical distance data between ripe and unripe fruits are available in the Source Data File and published open access on the Zenodo repository, https://doi.org/10.5281/zenodo.2634368. Note that behavioural data for all fruit species were used to generate Supplementary Fig. 1. Data for fruit species accompanied by fruit trait data were used to generate Fig. 1–4 and Supplementary Figs. 2–3.

## Code availability
R codes are posted on GitHub: https://github.com/omernevo/Variation-in-capuchin-sniffing-project.

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

## Acknowledgements

We thank R. Blanco Segura and M. M. Chavarria, and staff from the Área de Conservación Guanacaste and Ministerio de Ambiente y Energía, and Linda Fedigan and Jeremy Hogan from the University of Calgary for access to the P.A.C.E. database. Warmest thanks also to A. Guadamuz, A. Blauel, B. Klug, M. Lemmon, N. Parr, C. Sendall and L. Weckman for assistance with data collection. Funding was provided by The Leakey Foundation, the Canada Research Chairs programme and the National Sciences and Engineering Research Council of Canada (A.D.M.), and the Japan Society for the Promotion of Science 15H02421, 16K14818 and 18H04005 (S.K.). O.N was funded by the German Science Foundation (Deutsche Forschungsgemeinschaft—DFG, grant number NE 2156/1-1) during work on this manuscript. K.T. and M.S. was funded by ERATO Touhara Chemosensory Signal Project (JPMJER1202) and JST-Mirai Programme (JPMJMI17DC) from JST, Japan.

## Author contributions

A.D.M. and S.K. conceived the study. A.D.M, S.K., O.N. and K.T. designed the research. A.D.M., M.S. M.E, Y.M., K.S. and R.E.W. performed research. O.N., A.D.M., M.S. and E.G. analysed data. A.D.M. and O.N. wrote the initial paper. S.K., S.M. and K.T. contributed to subsequent writing. All authors read and approved the final version of the manuscript.
