## [Peer Review File · Nature Communications]

Reviewer #1 (Remarks to the Author):

In the manuscript "Fruit scent and observer colour vision shape food-selection strategies in wild capuchin monkeys", Melin et al. test three main predictions: 1) that the volume of fruit odorants is correlated with the frequency of sniffing behavior; 2) that colour vision phenotype is related to sniffing behavior; and 3) that plants that invest in visual signaling do so at the expense of olfactory signaling. The statistical analyses are all appropriate. The use of GLMM and the tests of model assumptions are all appropriate to account for repeated measures on the same individuals.

Overall, the manuscript is an interesting approach to understanding evolutionary trade-offs from the perspective of both the predator (monkey) and the prey (plants). This novel study utilizes an appropriate study system, New World monkeys, they have variation in colour-vision phenotypes within the social group, providing a wonderful system to test their predictions. This manuscript and study were well thought out and organized, and should be published with minor revisions. I find this unique and multidimensional approach of combining genetics, behavior, and fruit chemical compounds extremely integrative and I am excited about the prospects for our greater understanding of diet selection in frugivorous mammals.

A few issues need to be addressed. It is not clear why only 18 out of the 83 plant species were included in the analyses for colour vision and scent. This seems like a small percentage of the actual capuchin diet and thus limits the scope of analysis for this study – surely in 25 months there were more than 18 species that would qualify for both measures?

On line 142-141, the authors state that 91% of the variance in sniffing was explained and cite Table 1 – yet I do not see this in Table 1. This information should be included in the Table legend. It is also interesting that chemical odorants did not differ between ripe and unripe fruits in terms of sniffing behavior (using the Bray-Curtis dissimilarity index method) but the scent increase ratio did vary with sniffing behavior. Why is the chemical dissimilarity index called qualitative in the methods (line 324)? This should be explained more clearly in the discussion.

My main concern is how the authors can rule out sex as a factor that could be confounded within the colour-vision phenotype? For example, it has been documented that in humans, females have a better sense of smell relative to men. Given that sex is not included as a variable in the model, isn't it the case that the observed variation could be a consequence of sex? In this study system, males are all dichromats and females are both dichromats and trichromats. Is the dataset large enough such that the analysis can be run just for females? Or sex could be included in the model. Maybe males have a more acute sense of smell than females and hence dichromats smell more than females? Given that this pattern held irrespective of luminance contrast and red-green, blue-yellow fruits (lines 234-235), perhaps sex is the reason and not colour- vision phenotype. This does not seem to be addressed in the manuscript.

The finding that they sniff fruits that have more odorants is not very surprising and expected, but also validates the odorants measures.

Line 213, what types of difference in foraging strategies? Provide a reference

Line 214-215: What is meant by often – this should be quantified.

There are a lot of studies available on humans and olfaction, and surely on colour vision. In lines 241-243 possibly could be substantiated with studies on humans – are there any examples of this trade-off in humans?

Line 317 – why was 20% set as the threshold for exclusion?

Reviewer #2 (Remarks to the Author):

These are very valuable and exciting data. Not only is olfactorily-related behaviour rarely recorded in wild primates, the link to vision phenotype is even more valuable. The methods seem robust, the sample sizes large, and the analyses sound. The writing is very lucid. My comments are minor ones, readily dealt with.

Figure 1: the legend says “Proportions and standard deviation” but these are boxplots, so more likely medians, upper/lower quartiles (the box), with whiskers stretching to the first data point within 1.5 inter-quartile ranges of the box, and points beyond the whiskers as individual symbols. The legend needs to provide this information, because boxplot formats do vary.

Figure 2: the legend needs to explain what’s plotted – it looks like a boxplot overlaid on the raw data, jittered horizontally.

Figure 3. Presumably the dots are randomly jittered within each of the four quadrants? It's also not clear what the line under the "glmm: $p < 0.01$ " indicates (if anything). Dichromat/trichromat is a binary predictor so there can be no fitted line, as such, just two predicted proportions (one for each visual system).

Supplementary material:

Figure 1: same point as figure 1 in the main paper.

Figures 2 & 3: same point as figure 3 in the main paper.

Supplementary Note 2: Colour Contrast Models: what is the role of the "3100 K tungsten halogen calibration light source" in measuring irradiance? Presumably none, so this is a mistake and the light source should be mentioned only when discussing measurement of reflectance spectra.

Reviewer #3 (Remarks to the Author):

I was asked to review the chemical ecology part (VOC sampling and analysis) of this paper. First of all, I think the methods should be described in full in the main body of the text, not just in the supplementary. Several things are not clear in the descriptions of the methods. What is a MonoTrap RGC18 adsorbent? Please state the kind of adsorbent that was used in this trap (Tenax, Super Q, etc.)? Were there any pumps used or simply passive absorbance? What was the flow rate, etc.? For the VOC analysis, as stated in the text, comparison of MS spectra with NIST library spectra is clearly not sufficient for volatile identification. An "above 85%" confidence criterion will lead to several wrongly identified compounds. The authors need to justify why this is not a problem in their paper. Also problematic is the method used for VOC quantification. The authors mention they used the peak area. The peak area (I assume of total ion chromatograms), however, varies in its association to the amount of a compound, because different compound classes produce different amounts of ions in a mass spectrometer. Therefore, people usually use some sort of calibration with dose-response curves, with e.g. one representative compound for all compound classes included in the study.

Reviewers' comments:

Reviewer #1 (Remarks to the Author):

In the manuscript “Fruit scent and observer colour vision shape food-selection strategies in wild capuchin monkeys”, Melin et al. test three main predictions: 1) that the volume of fruit odorants is correlated with the frequency of sniffing behavior; 2) that colour vision phenotype is related to sniffing behavior; and 3) that plants that invest in visual signaling do so at the expense of olfactory signaling. The statistical analyses are all appropriate. The use of GLMM and the tests of model assumptions are all appropriate to account for repeated measures on the same individuals.

Overall, the manuscript is an interesting approach to understanding evolutionary trade-offs from the perspective of both the predator (monkey) and the prey (plants). This novel study utilizes an appropriate study system, New World monkeys, they have variation in colour-vision phenotypes within the social group, providing a wonderful system to test their predictions. This manuscript and study were well thought out and organized, and should be published with minor revisions. I find this unique and multidimensional approach of combining genetics, behavior, and fruit chemical compounds extremely integrative and I am excited about the prospects for our greater understanding of diet selection in frugivorous mammals.

We thank the reviewer for the positive comments and for the numerous, useful suggestions that have improved our manuscript.

A few issues need to be addressed. It is not clear why only 18 out of the 83 plant species were included in the analyses for colour vision and scent. This seems like a small percentage of the actual capuchin diet and thus limits the scope of analysis for this study – surely in 25 months there were more than 18 species that would qualify for both measures?

Thank you for highlighting the need for us to explain this. This choice was primarily based on financial and time (analytical) limitations, as well as access to fruits across ripeness

categories. This project was quite expensive and analytically demanding. Therefore, we strategically chose a diverse selection of important dietary fruit species. We have now clarified our justification in the text.

Lines 298-301 “The choice to sample a subset was based on financial limitations and access to fruits across ripeness categories. We were strategic in our selection, choosing fruits spanning different colour categories, and including fruit species for which monkey varied in their propensity for olfactory investigation, ranging from never sniffed to often sniffed.”

On line 142-141, the authors state that 91% of the variance in sniffing was explained and cite Table 1 – yet I do not see this in Table 1. This information should be included in the Table legend.

Thank you. We have now added this to the table legend as requested.

“Table 1. Results of generalized linear mixed model. Coefficients and p-values of all fixed effects. This model explained 91% of the variance in fruit sniffing behaviour by wild white-faced capuchin monkeys; 45% of the variance was explained by the fixed factors shown in the table, and 46% by the random factors (individual, plant species).”

It is also interesting that chemical odorants did not differ between ripe and unripe fruits in terms of sniffing behavior (using the Bray-Curtis dissimilarity index method) but the scent increase ratio did vary with sniffing behavior. Why is the chemical dissimilarity index called qualitative in the methods (line 324)? This should be explained more clearly in the discussion.

Thank you for highlighting the need for clarification. To avoid introducing new terms that only appear twice in the methods section and raise confusion, we decided to remove reference to “quantitative” and “qualitative” measures. (The ratio of odorant differences between two materials is sometimes called qualitative because it ignores the absolute amounts of the odourants). We feel the manuscript is more accessible if we drop these terms.

To clarify our discussion of these results, we also revised the second paragraph of our discussion and added a few clarifying sentences.

Lines 200-205 “Lacking an a-priori knowledge which VOCs are used to identify ripeness, our measurement of ripe-unripe dissimilarity (Bray-Curtis dissimilarity index) included all VOCs in the dataset. Given that some of those are likely to play other roles such as defence, this inclusive approach may have introduced noise (variation in compounds irrelevant to capuchins) that hindered us from detecting the effect of changes in chemical composition that are meaningful to capuchins.”

My main concern is how the authors can rule out sex as a factor that could be confounded within the colour-vision phenotype? For example, it has been documented that in humans, females have a better sense of smell relative to men. Given that sex is not included as a variable in the model, isn't it the case that the observed variation could be a consequence of sex? In this study system, males are all dichromats and females are both dichromats and trichromats. Is the dataset large enough such that the analysis can be run just for females? Or sex could be included in the model. Maybe males have a more acute sense of smell than females and hence dichromats smell more than females? Given that this pattern held irrespective of luminance contrast and red-green, blue-yellow fruits (lines 234-235), perhaps sex is the reason and not colour-vision phenotype. This does not seem to be addressed in the manuscript.

We thank the reviewer for raising this important point. We have now addressed the possible confounding effect of sex in two ways:

1) We followed the reviewer's suggestion to run the model on females only. Running the full model with this decreased sample size yielded essentially the same results; trichromats are significantly less likely to sniff fruits than dichromats are (color vision $p < 0.01$), and an increase in scent increase ratio is positively associated with sniffing behavior. The slopes (=effect sizes) are also very similar. However, a convergence warning occurred when running the full model. Therefore, to be certain our results held, we ran a less complicated model (removed fruit color classifications, which were not significant). This simpler model converged, and the results remained consistent. This further supports our conclusions that color vision type influences olfactory behaviors.

2) Additionally, to explicitly test the impact of sex, we ran the model again on dichromats only (males and females), and included sex as a variable. As previously, an increase in scent production with ripening is positively associated with sniffing behaviour ($p < 0.01$). However, we also revealed an impact of sex, when color vision is removed. Dichromatic females sniffed fruits significantly more than males did. This further strengthens our results: i.e. trichromatic females sniff fruits less often, even though females overall seem to use their sense of smell more than males.

We now report the results of these two models in the results section and provide full results in the supplementary information (Supple. Table X and Y). We also include a statement of these tests in our methods. We report the results of the simpler models that converged well (Supplementary Table 1,2). Results of the full and simplified models can be found in the source code.

Results:

Lines 155-165: “Colour vision phenotype had a significant impact on sniffing behaviour. Controlling for fruit species and individual ID, dichromatic monkeys sniffed fruits more often than trichromatic monkeys did (38.8% versus 28.6% of the data points, respectively; $p < 0.01$; Table 1; Fig. 3). Because sex and colour vision type are confounded in this system, we also ran models for females only. We achieved comparable results, i.e. a significant effect of colour vision type (Supplementary Table 1). The impact of colour vision phenotype held irrespective of the red-green, blue-yellow or luminance contrast between ripe and unripe fruits; fruits ranging from high to low visual conspicuity along each of these dimensions were found in both the sniffed and non-sniffed categories (Fig. 3, Supplementary Fig. 2, 3). When we removed trichromatic females from the analysis to examine the impact of sex alone, we found that dichromatic males sniffed fruits significantly less than dichromatic females (Supplementary Table 2).”

Methods:

Lines 365-374: “We ran two additional models to examine the potentially confounding effects of sex and colour vision type. 1) We first ran a model that included only females in the analyses to assess colour vision type in the absence of male data; 2) We then ran a model that included only dichromats (removing trichromatic females) to test for an effect of sex. The lower sample sizes of the latter two models caused convergence warnings. To remedy this, we simplified the model by removing the fruit colour categories, which successfully resolved the warning message. Importantly, the results were essentially the same whether or not fruit colour was included, i.e. no change in which variables had a significant effect. We report the results of the simpler models that converged well (Supplementary Table 1,2). Results of the full and simplified models can be found in the source code.”

Line 213, what types of difference in foraging strategies? Provide a reference

Our paper on this has not yet been published, but we now cite the abstract reporting the species-specific differences.

Line 214-215: What is meant by often – this should be quantified.

Thank you for this suggestion. We have now added the data to back up our statement.

Lines 223-225: consistent with⁴⁵ we find that capuchins sniff visually cryptic, ever-green figs (*Ficus morazaniana*) relatively often (ca. 40% of the time; Supplementary Fig. 1).

There are a lot of studies available on humans and olfaction, and surely on colour vision. In lines 241-243 possibly could be substantiated with studies on humans – are there any examples of this trade-off in humans?

We conducted an extensive literature review and unfortunately there is an absence of research on use of olfaction in colorblind versus “color-normal” humans. However, we have now inserted information on this trade-off in blind versus sighted subjects.

Lines 252-262: Data on olfactory threshold sensitivity and cognitive correlates in visually-impaired humans are mixed. Blind subjects have been found to have lower olfactory sensitivity thresholds, larger olfactory bulbs, and to rely more on their sense of smell than sighted subjects⁵⁹. For example, compared to matched-age sighted children, visually-impaired children used olfaction more often to glean social information and paid more attention to odours of unfamiliar foods⁶⁰. However, a recent meta-analysis concluded that blindness seems not to affect odour identification, odour discrimination or odour thresholds systematically. Importantly, however, this review assessed olfactory performance and not relative use of the sense of smell during food selection or facets of daily life. Further work exploring the development of sensory biases and neurological correlates among primates with different colour vision types would be instructive⁶¹.”

Line 317 – why was 20% set as the threshold for exclusion?

The 20% cutoff line is indeed somewhat arbitrary. It was decided based on our prior experience dealing with these types of samples and was aimed at offering a good balance between not being too conservative (i.e. excluding compounds that may be relevant) or permissive (i.e. including compounds that are likely to be contaminants and distort the results). Our rationale is that a compound is unlikely to be a genuine part of the species' scent profile, or at least of interest to animals feeding on the fruit, if it's very uncommon within the species. On the other hand, if a compound is present consistently across samples, it is less likely to be a contaminant and therefore more likely to be of biological relevance. It is important to note that the excluded compounds tended to be of very small amounts. This means that their effect on the indices we calculated and used for our analysis is miniscule. We now more thoroughly explain our justification through adding an extra paragraph to Supplementary Note 4:

“To reduce the amount of noise and focus on more relevant VOCs, we also excluded all very rare compounds, defined as those identified in less than 20% of the samples in each species. Our rationale is that if a compound is consistently present across samples, it is less likely to be a contaminant and more likely to be of biological relevance. The cut-off of 20%, while somewhat arbitrary, serves to strike a balance between erring on the side of being too conservative, i.e. excluding compounds which may be biologically relevant, or on the side of being too permissive, i.e. including compounds that are likely to be contaminants or present in biologically insignificant amounts. Importantly, the compounds that were excluded were present in very small amounts meaning that their effect, if included, on the indices we calculated and used for our analysis would have been extremely small.”

Reviewer #2 (Remarks to the Author):

These are very valuable and exciting data. Not only is olfactorily-related behaviour rarely recorded in wild primates, the link to vision phenotype is even more valuable. The methods seem robust, the sample sizes large, and the analyses sound. The writing is very lucid. My comments are minor ones, readily dealt with.

Thank you- We very much appreciate the positive and supportive comments and are pleased you find our data valuable.

Figure 1: the legend says “Proportions and standard deviation” but these are boxplots, so more likely medians, upper/lower quartiles (the box), with whiskers stretching to the first data point within 1.5 inter-quartile ranges of the box, and points beyond the whiskers as individual symbols. The legend needs to provide this information, because boxplot formats do vary.

Response: Thank you for alerting us to our error. You are correct about this and we have edited the figure legend accordingly, as follows:

“Figure 1. Variation in fruit sniffing behaviour by white-faced capuchins for 18 plant species. Box plots of fruit investigation sequences (N = 15 160) by individual white-faced capuchin monkeys that included at least 1 sniffing event. Medians (bar) are plotted along with upper and lower quartiles (box perimeters), and whiskers stretching to the first data point within 1.5 inter-quartile ranges of the box. Points beyond the whiskers are plotted as individual symbols.”

Figure 2: the legend needs to explain what’s plotted – it looks like a boxplot overlaid on the raw data, jittered horizontally.

You are correct. We have now added this explanation that our boxplots are overlaid on the raw data, jittered horizontally.

Figure 3. Presumably the dots are randomly jittered within each of the four quadrants? It’s also not clear what the line under the “glm: $p < 0.01$ ” indicates (if anything). Dichromat/trichromat is a binary predictor so there can be no fitted line, as such, just two predicted proportions (one for each visual system).

Thank you for this comment. We have edited the legend to indicate that data points are vertically and horizontally jittered in each quadrant.

The curve is the logistic regression model, predicting the probability of scoring either 1 or 0 (sniff or no sniff). We agree that the curve is not ideal because there are no intermediate

color-vision phenotypes. We have removed the curve and now indicate the result (glimm: $p < 0.01$) as part of the figure legend.

Supplementary material:

Figure 1: same point as figure 1 in the main paper.

We have revised this accordingly

Figures 2 & 3: same point as figure 3 in the main paper.

We have revised this accordingly

Supplementary Note 2: Colour Contrast Models: what is the role of the “3100 K tungsten halogen calibration light source” in measuring irradiance? Presumably none, so this is a mistake and the light source should be mentioned only when discussing measurement of reflectance spectra.

Thank you for alerting us to this source of confusion. The light source was initially used to calibrate the spectrometer, and is not used during measurement of irradiance. We have now clarified this in the text. “...using a USB2000 spectrometer that was calibrated with a 3100 K tungsten halogen calibration light source (LS-1-CAL, Ocean Optics)¹.”

Reviewer #3 (Remarks to the Author):

I was asked to review the chemical ecology part (VOC sampling and analysis) of this paper. First of all, I think the methods should be described in full in the main body of the text, not just in the supplementary.

Thank you - we understand the motivation for this recommendation and agree this would be ideal. However, because we are limited by word count and we prefer to provide a very thorough accounting of our methods (and additionally appreciate that these details might be a bit cumbersome for non-specialists), with respect, we feel that use of Supplementary Information is a more appropriate space for this. We would be willing to revise this position if the editor agrees with Reviewer 3 and favours moving this information to the main text.

Several things are not clear in the descriptions of the methods. What is a MonoTrap RGC18 adsorbent? Please state the kind of adsorbent that was used in this trap (Tenax, Super Q, etc.)? Were there any pumps used or simply passive absorbance? What was the flow rate, etc.?

Thank you for highlighting these areas needing further clarification. We apologize for omitting some of these important details and have now added them to the methods section, as follows:

Lines 312-314: “To quantify fruit scent, we sampled the VOCs passively (without a pump) from the fruits using MonoTrap RGC18 TD rods (GL Sciences Inc). These are portable scent traps designed for strong passive absorbance based on a graphite carbon and octa-decyl-silyl composition.”

For the VOC analysis, as stated in the text, comparison of MS spectra with NIST library spectra is clearly not sufficient for volatile identification. An “above 85%” confidence criterion will lead to several wrongly identified compounds. The authors need to justify why this is not a problem in their paper.

Thank you for raising this point. We fully agree that this not sufficient for accurate volatile identification, and therefore make it clear in the text that the ID assigned is tentative (line 330). Because of this, we used the IDs only to identify and remove common contaminants such as phthalates and siloxanes, which are easy to identify based only on mass spectra.

Crucially, the tentativeness of the ID - and possibility that some peaks were assigned to the wrong compound *name* - does not impact our results. For each fruit species, we aligned the peaks so that a compound always got the same ID across samples. This means that compound ID has no effect on the distance metric (Bray-Curtis dissimilarity index) we calculated, and hence does not impact the results.

We now clarify these points in the main text:

Lines 334-337: Importantly, the limitations in accurately identifying the compound names does not impact our results or conclusions. For each fruit species, we aligned the peaks so that a compound was always the same ID across samples. This means that the compound ID itself has no effect on the distance metric we calculated, and hence does not influence the results.

Finally, we also wish to mention that, unfortunately, ID assignment based on analytical standards is not realistic for research of this scale, in which hundreds of different VOCs are analyzed and some are difficult to identify, and perhaps unknown. Our manuscript therefore focuses on more robust indices of chemical profiles that we could calculate and does not discuss the actual VOCs identified.

Also problematic is the method used for VOC quantification. The authors mention they used the peak area. The peak area (I assume of total ion chromatograms), however, varies in its association to the amount of a compound, because different compound classes produce different amounts of ions in a mass spectrometer. Therefore, people usually use some sort of calibration with dose-response curves, with e.g. one representative compound for all compound classes included in the study.

We agree with the reviewer that using dose-response curves for the compounds would increase the accuracy of the calculations. This was not practical because of our very large and diverse dataset, which included hundreds of different VOCs across 18 fruit species, each of two ripeness states.

However, we believe that the effect of this is minimal and that our results are valid for our goals because our index does not aim to assess absolute amounts, but rather to calculate the ratio between the overall amount - indeed based on the total ion chromatograms (TIC) - in ripe and unripe fruits within the same species. The chemicals present in ripe and unripe fruits of a given species tend to come from the same VOC classes (e.g. terpenoids): this is apparent in the present study and in a forthcoming manuscript from another system, where one of the coauthors (ON) found a strong positive correlation (correlation coefficients 0.75, 0.66, 0.68 for aliphatics, aromatics and terpenoids - the main plant VOC classes) between the share of various chemical classes in ripe and unripe fruits of the same species. Therefore, any calibration factor would be similar in ripe and unripe fruits and be negated to a large extent when calculating the ratio. While we agree that other methods would be more appropriate for assessing absolute amounts, we believe that our index is an adequate approximation for the difference in overall VOC emission between ripe and unripe fruits of a single species.

Finally, we note that using GC-MS TIC as a proxy for overall amount of scent has been used in recent studies with simpler datasets than ours (e.g. Krug et al. 2018, *Frontiers in Plant Science*; Chapurlat et al. 2019, *New Phytologist*). We therefore believe that given the complexity of our data and the fact that we do not attempt to calculate absolute amounts, but instead the intraspecific ratio between fruits of differing ripeness, our approach is within the standards of the field.

We have now more completely disclosed and discussed this limitation in our manuscript by adding the following paragraph to Supplementary Note 5:

“A limitation of our study is that the accuracy of quantifying VOC TIC area would have been improved by using dose-response curves for the different compound classes. Unfortunately, this was not practical because of the very large and diverse dataset, which included hundreds of different VOCs across 18 fruit species, each of two ripeness states. However, we believe that the effect of this is minimal because our index does not aim to assess absolute amounts, but rather to calculate the ratio between the overall amount in ripe and unripe fruits within species. Because the chemicals present in ripe and unripe fruits of a given species tend to come from the same VOC classes, any calibration factor would be negated to a large extent when calculating the ratio. Therefore, the index we use is a reasonable approximation for the difference in overall VOC emission between ripe and unripe fruits of a single species.”

Reviewer #1 (Remarks to the Author):

In the manuscript "Fruit scent and observer colour vision shape food-selection strategies in wild capuchin monkeys", Melin et al. test three main predictions: 1) that the volume of fruit odorants is correlated with the frequency of sniffing behavior; 2) that colour vision phenotype is related to sniffing behavior; and 3) that plants that invest in visual signaling do so at the expense of olfactory signaling. The statistical analyses are all appropriate. The use of GLMM and the tests of model assumptions are all appropriate to account for repeated measures on the same individuals.

Overall, the manuscript is an interesting approach to understanding evolutionary trade-offs from the perspective of both the predator (monkey) and the prey (plants). This novel study utilizes an appropriate study system, new world monkeys, they have variation in colour-vision phenotypes within the social group, providing a wonderful system to test their predictions. This manuscript and study were well thought out and organized, and should be published.. I find this unique and multidimensional approach of combining genetics, behavior, and fruit chemical compounds extremely integrative and I am excited about the prospects for our greater understanding of diet selection in frugivorous mammals.

I am very satisfied with the revisions to the manuscript and feel that the authors have greatly strengthened the manuscript. The authors addressed all of my concerns, conducted the new analyses to test for the affect of sex, and I think the results of these new analyses are highly convincing. I will defer to the other reviewer about the revisions made regarding the questions on odor and chemical composition of fruit odors.

Reviewer #3 (Remarks to the Author):

I think the methods are - within the given constraints - adequately described now.

REVIEWERS' COMMENTS:

Reviewer #1 (Remarks to the Author):

In the manuscript "Fruit scent and observer colour vision shape food-selection strategies in wild capuchin monkeys", Melin et al. test three main predictions: 1) that the volume of fruit odorants is correlated with the frequency of sniffing behavior; 2) that colour vision phenotype is related to sniffing behavior; and 3) that plants that invest in visual signaling do so at the expense of olfactory signaling. The statistical analyses are all appropriate. The use of GLMM and the tests of model assumptions are all appropriate to account for repeated measures on the same individuals.

Overall, the manuscript is an interesting approach to understanding evolutionary trade-offs from the perspective of both the predator (monkey) and the prey (plants). This novel study utilizes an appropriate study system, new world monkeys, they have variation in colour-vision phenotypes within the social group, providing a wonderful system to test their predictions. This manuscript and study were well thought out and organized, and should be published.. I find this unique and multidimensional approach of combining genetics, behavior, and fruit chemical compounds extremely integrative and I am excited about the prospects for our greater understanding of diet selection in frugivorous mammals.

I am very satisfied with the revisions to the manuscript and feel that the authors have greatly strengthened the manuscript. The authors addressed all of my concerns, conducted the new analyses to test for the affect of sex, and I think the results of these new analyses are highly convincing. I will defer to the other reviewer about the revisions made regarding the questions on odor and chemical composition of fruit odors.

We very much appreciate the positive comments and thank Reviewer 1 for numerous, helpful suggestions on the previous version of our manuscript.

Reviewer #3 (Remarks to the Author):

I think the methods are - within the given constraints - adequately described now.

We are glad our revisions were satisfactory and thank you for your helpful feedback on previous versions of our manuscript.